# Multi-Branch of Attention Yields Accurate Results for Tabular Data

## Abstract

Tabular data inherently exhibits significant feature heterogeneity, but existing transformer-based methods lack specialized mechanisms to handle this property. To bridge the gap, we propose MAYA, an encoder-decoder transformer-based framework. In the encoder, we design a **M**ulti-**B**ranch of **A**ttention (MBA) that constructs multiple parallel attention branches and averages the features at each branch, effectively fusing heterogeneous features while limiting parameter growth. Additionally, we employ collaborative learning with a dynamic consistency weight constraint to produce more robust representations. In the decoder, cross-attention is utilized to seamlessly integrate tabular data with corresponding label features. This dual-attention mechanism effectively captures both intra-instance and inter-instance interactions. We evaluate the proposed method on a wide range of datasets and compare it with other state-of-the-art transformer-based methods. Extensive experiments demonstrate that our model achieves superior performance among transformer-based methods in both tabular classification and regression tasks.

## 1 Introduction

As society advances rapidly, tabular data have become a widely used format (Zhou et al., 2025). Despite its complex and multifaceted nature, its significant economic value drives growing researcher interest in analysis and study, leading to the proposal of various solutions (Klambauer et al., 2017; Popov et al., 2020; Wang et al., 2021; Arik & Pfister, 2021; Bonet et al., 2024; Ye et al., 2024).

Inspired by transformers' success in NLP and speech, transformer-based architectures are increasingly adopted for tabular data. AutoInt (Song et al., 2019) projects numerical and categorical features into a shared low-dimensional space and employs a multi-head self-attention network with residual connections to model feature interactions, enabling efficient end-to-end training on large-scale raw data. To investigate the effectiveness of embeddings, Huang et al. (2020) propose transforming the embeddings of categorical features into robust contextual representations, achieving performance comparable to tree-based methods (Chen & Guestrin, 2016; Prokhorenkova et al., 2018) on some datasets. Then, FT-Transformer (Gorishniy et al., 2021) achieves strong performance across multiple datasets, demonstrating potential for superiority. Afterwards, ExcelFormer (Chen et al., 2024) introduces a semi-permeable attention module that selectively lets informative features integrate data from less relevant ones. To further improve performance, DANets (Chen et al., 2022) and BiSHop (Xu et al., 2024) enhance model competitiveness by developing feature interaction modules (intra- and inter-instance) to boost performance. Moreover, AMFormer (Cheng et al., 2024) enhances feature representation through arithmetic operations beyond basic interactions.

Current transformer architectures struggle to effectively process heterogeneous data (Chen et al., 2023; 2024; Cheng et al., 2024). While diverse feature extraction is promising, simply increasing attention heads is inefficient. This approach leads to an expansion of feature dimensions due to the concatenation operation, consequently increasing the parameters quadratically of subsequent Feed-Forward Network (FFN) layers and compromising computational efficiency. Thus, a transformer architecture for effective diverse feature extraction from tabular data is crucial.

In this work, we propose MAYA (**M**ixture of **A**ttention **Y**ields **A**ccurate results for tabular data), a novel encoder-decoder transformer-based framework that achieves rich feature representations while maintaining parameter efficiency. Specifically, in the encoder, we propose an innovative

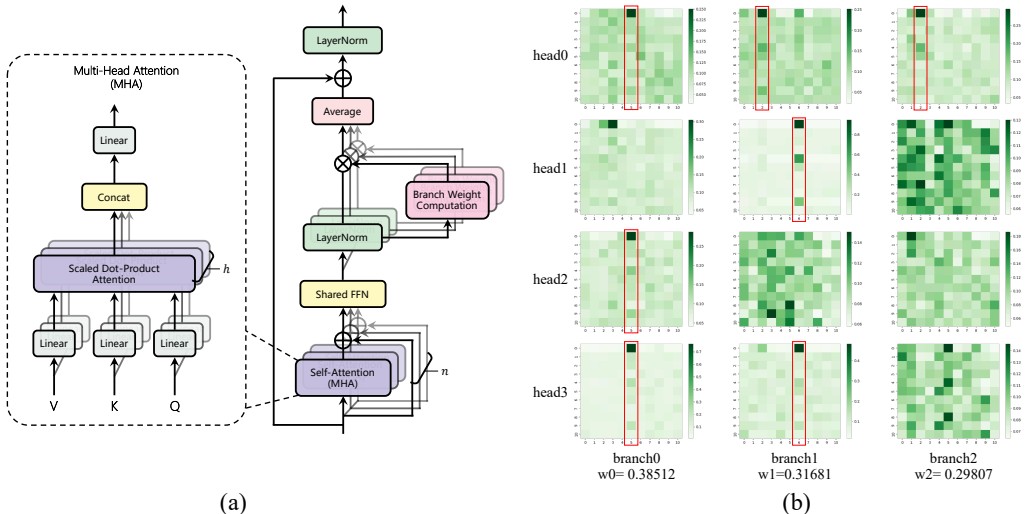

(a)                    (b)

Figure 1: (a) The overview of MBA block. It employs multiple parallel MHA branches to enrich feature extraction. To maintain a constant size of hidden states, features are averaged branch-wise with dynamic weights, imposing a consistency constraint and enabling collaborative branch learning. (b) The visualization of the MBA block. Each branch exhibits distinct modes of feature interaction, such as sparse, dense, and hybrid patterns.

**M**ulti-**B**ranch of **A**ttention (MBA) structure, as illustrated in Fig.1(a), which implements parallelized independent attention branches within a single encoder block. This architecture enables effective feature fusion while significantly improving feature diversity for tabular data. Inspired by collaborative mechanisms (Ke et al., 2020), our framework weights multi-branch attention outputs via collaborative policy for robust representations. In addition, we cascade multiple MBA blocks, enabling the capture of high-level semantic information and enhancing the encoder's representation capacity. The MBA's multi-branch design achieves two key objectives: it extracts diverse features while controlling dimensionality growth through averaging operations, thereby substantially reducing the parameter burden on downstream FFN layers. Unlike traditional concatenation, this avoids quadratic parameter growth in later layers. Furthermore, the decoder's inter-instance attention and label integration yield more reliable results.

MAYA outperforms SOTA tabular transformers across multiple datasets. Furthermore, ablation studies confirm MBA's effectiveness, positioning MAYA as a new tabular transformer baseline. The main contributions of MAYA are:

- We empirically verify on several public datasets that the proposed MBA structure can effectively obtain diverse features to deal with heterogeneity in tabular data.

- We show that the collaborative mechanism applied in branch weight computation of MBA can improve consistency training for tabular data.

- We propose a transformer encoder-decoder with specialized attention mechanisms for intra- and inter-instance relationships.

## 2 RELATED WORK

### 2.1 ATTENTION MECHANISMS

The attention mechanism was introduced to improve the performance of the encoder-decoder model for machine translation (Vaswani, 2017). It reveals the relative importance of each component in a sequence compared to the other components. Usually, self-attention and cross-attention are the most widely used in transformer architecture models. Self-attention captures relationships within a single input sequence, and cross-attention captures relationships between elements of different input

sequences. Somepalli et al. (2021) introduce self-attention and inter-sample attention mechanisms for the classification of tabular data. They use self-attention for intra-sample features and inter-sample attention for inter-sample relationships. In our approach, we introduce two novel attention mechanisms: the Multi-branch of Attention (MBA) for self-attention and Inter-instance Attention Incorporating with Labels (IAIL) for cross-attention, to enhance data utilization and feature representation. Compared to Somepalli et al. (2021), the inter-instance attention mechanism of IAIL is more parameter-efficient.

## 2.2 COLLABORATIVE LEARNING

Collaborative learning (Dillenbourg, 1999) refers to methods and environments in which learners work together on a shared task, with each individual relying on and being responsible to others. In pixel-wise tasks, collaborative policy has been used to train different subset networks and to ensemble the reliable pixels predicted by these networks. As a result, this strategy covers a broad spectrum of pixel-wise tasks without requiring structural adaptation (Ke et al., 2020). Here, we use collaborative policy to construct branch weight in MBA block during training, with the weight proportional to the difference between the branch output and the true label. This dynamic consistency constraint can provide substantial benefit to enrich feature representations.

## 3 METHODS

In this section, we present a detailed introduction to the proposed MAYA, adopting a top-down approach—starting with the network's holistic view and progressing to its fundamental components.

## 3.1 NOTATIONS

In this work, let's consider supervised learning problems on tabular data. A typical dataset with $N$ instances can be denoted as $\mathcal{D} = \{(x_i, y_i)\}_{i=1}^{N}$, where $x_i$ represents the features and $y_i$ the label of the $i$-th instance, respectively. Specifically, $x_i$ contains both numerical features $x_i^{num}$ and categorical features $x_i^{cat}$, that is to say, $x_i = (x_i^{num}, x_i^{cat})$. To be convenient and without loss of generality, we denote an instance with $k$ features as $x_i \in \mathbb{R}^k$. For the label $y_i$, three types of tasks are considered, which are binary classification $y_i \in \{0, 1\}$, multi-class classification $y_i \in \{1, \ldots, C\}$ and regression $y_i \in \mathbb{R}$.

## 3.2 TOKENIZER

Built upon transformer-like architecture, MAYA needs a tokenizer to map instances into an embedding space. For an instance $x_i \in \mathbb{R}^k$, we define the $d$-dimension embeddings obtained by tokenizer as $z_i$:

$$z_i = \text{Tokenizer}(x_i), z_i \in \mathbb{R}^{k \times d} \tag{1}$$

The tokenizer can be implemented in various ways, while in this paper, we primarily utilize the tokenizer of Gorishniy et al. (2021), which embeds numerical features through element-wise multiplication and categorical features via a lookup table.

Unlike Gorishniy et al. (2021), we innovatively find that adding an activation function after the tokenizer boosts network performance. We attribute this to the Kaiming initialization of tokenizer weights/biases which pairs effectively with ReLU. We'll explore this detail in Section 4.3 via ablation study.

Before the extraction of features, we append the embedding of a special token, i.e. [CLS] token, to $z_i$:

$$z_i^0 = \text{Stack}[[CLS], z_i], z_i^0 \in \mathbb{R}^{(k+1) \times d} \tag{2}$$

where the subscript "0" denotes the initial input, i.e., the input to the first layer of the entire network.

## 3.3 ENCODER-DECODER ARCHITECTURE

Our network adopts an encoder-decoder framework (Fig.2). The encoder processes the initial input $z_i^0$ into $\hat{z}_i$ (i.e. the processed [CLS] token representing the instance). $\hat{z}_i$ is then fed into the decoder

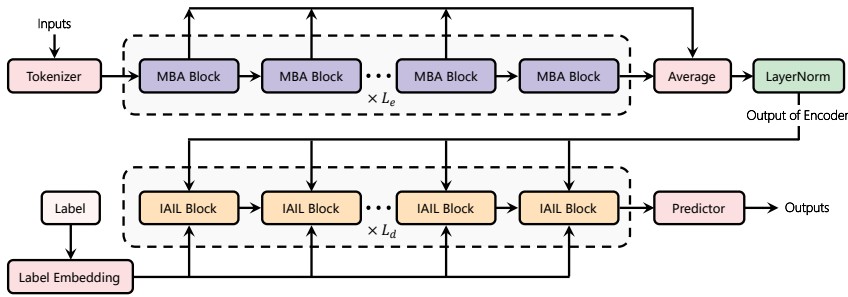

Figure 2: The overall structure of MAYA. It comprises an encoder (with MBA blocks) and a decoder (with IAIL blocks), arranged in upper and lower rows respectively. Inputs are tokenized before encoder processing, and decoder's outputs are fed into a Predictor for final predictions.

to produce $\tilde{z}_i$, and a predictor generates the final prediction $p$. The process is summarized as:

$$\hat{z}_i = \text{Encoder}(z_i^0), \hat{z}_i \in \mathbb{R}^d \tag{3}$$

$$\tilde{z}_i = \text{Decoder}(\hat{z}_i), \tilde{z}_i \in \mathbb{R}^d \tag{4}$$

$$p = \text{Predictor}(\tilde{z}_i) \tag{5}$$

where the predictor is a simple fully-connected layer.

### 3.4 ENCODER

The construction of the encoder employs a deep-wide integration strategy: "deep" via traditional block cascading, and "wide" using our novel MBA block with parallel attentions. An encoder with $L_e$ MBA blocks is thus structured as:

$$z_i^\ell = \text{MBA}^\ell(z_i^{\ell-1}), \ell \in [1, L_e] \tag{6}$$

#### 3.4.1 MULTI-BRANCH OF ATTENTION (MBA)

The heterogeneity of tabular data makes a single feature extractor inadequate. The classical transformer's Multi-Head Attention (MHA) focuses on diverse representation subspaces via multiple heads (Vaswani, 2017), suggesting enhanced feature-processing capacity with more heads. However, this approach has two drawbacks: (1) concatenating MHA head subspaces enlarges attention output hidden states, thus increasing the number of parameters in FFN and computational cost; (2) output projection of concatenated features causes mixing, reducing subspace diversity.

To tackle the issues, we introduce the Multi-branch of Attention (MBA, Fig.1(a)), which employs multiple parallel MHA-based attention branches with branch-level feature averaging. This approach preserves feature extraction diversity and constant hidden state sizes, avoiding FFN parameter inflation. In addition, feature averaging serves as regularization, reducing overfitting risk and improving robustness, as well as mitigating the impact of individual MHA failures.

Specifically, the input is duplicated and fed into identical attention branches with different weights. Their outputs pass through a shared FFN and individual layer normalization, then are averaged and normalized by another layer normalization to form the MBA block's output. Formally, the output $z_i^\ell$ of the $\ell$-th MBA block (with $n$ parallel branches) is derived as follows:

$$z_i^\ell = \text{LayerNorm}(\frac{1}{n}\sum_{j=1}^{n}\text{Branch}_j(z_i^{\ell-1}) + z_i^{\ell-1}) \tag{7}$$

where $\text{Branch}_j(\cdot) := \text{LayerNorm}_j(\text{FFN}(\text{Attention}_j(\cdot) + (\cdot)))$.

#### 3.4.2 INTRA-BLOCK BALANCE: BRANCH WEIGHT OF MBA

The MBA block's output averages results from multiple attention branches, implying equal contribution from each branch—a collaborative learning paradigm. Then, we introduce a dynamic consistency constraint, proposing a balancing strategy via branch weights (Fig.3(a)). During training,

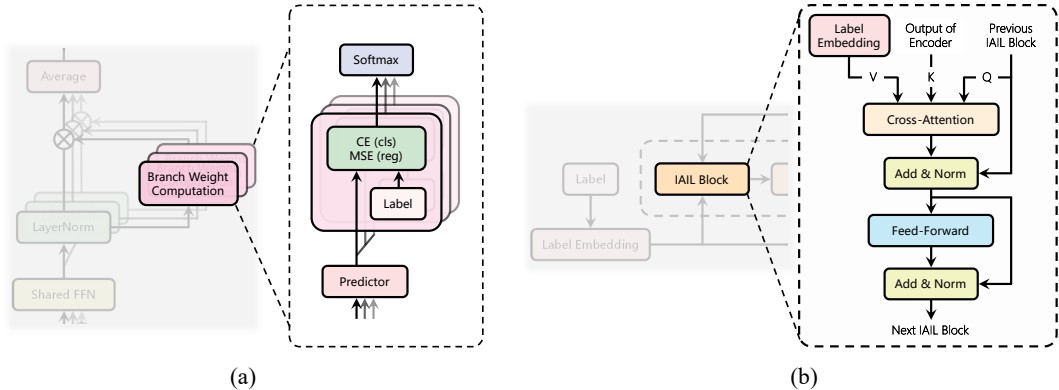

(a)                                      (b)

Figure 3: (a) Branch weights are computed during training by appending a shared Predictor to each branch, calculating losses, and assigning higher weights to branches with larger losses. This dynamic constraint promotes collaborative learning. (b) The overview of IAIL block in decoder.

we attach a shared Predictor (same as in Eq.5) to each attention branch. Branches with significant prediction deviations (quantified by MSE for regression or CE for classification) are assigned larger weights, which are computed via softmax over branch losses. This penalizes deviating branches, balancing their contributions. To stabilize the computation of branch weights, we apply Exponential Moving Average (EMA) for smoothing purposes. The branch weight $W_B$ is formulated as:

$$W_B = \text{EMA}(\text{Softmax}(s)) \tag{8}$$

where $W_B = \{w_j\}_{j=1}^n$ and $s = \{s_j\}_{j=1}^n$. $s_j$ is the supervised loss of $\text{Branch}_j$, that is

$$s_j = \text{L}_{pred}(\text{Predictor}(\text{Branch}_j(z^{\ell-1})), y) \tag{9}$$

where $\text{L}_{pred}$ is MSE loss or CE loss.

By applying the branch weight, Eq.7 can be reformulated as

$$z_i^\ell = \text{LayerNorm}(\sum_{j=1}^n w_j \text{Branch}_j(z_i^{\ell-1}) + z_i^{\ell-1}) \tag{10}$$

### 3.4.3 INTER-BLOCK AVERAGING

To balance both intra-block and inter-block feature extraction efficacy, we average the outputs of $L_e$ stacked MBA blocks, append a LayerNorm layer, and obtain the encoder's final output. Combining Eq.3 and Eq.6, the encoder's output $\hat{z}_i$ is derived as follows:

$$\hat{z}_i = \text{LayerNorm}(\frac{1}{L_e} \sum_{l=1}^{L_e} z_i^l) \tag{11}$$

### 3.4.4 ANALYSIS OF MBA

In this section, we present a more in-depth analysis and insights into the MBA block to validate its effectiveness. Fig.1(b) illustrates the attention maps of a specific layer within MAYA, where the MBA block comprises 3 branches, each with 4 attention heads. For details on the generation of Fig.1(b), please refer to the Appendix A. Given the utilization of the [CLS] token, in each attention map, features that significantly impact the [CLS] token are marked by correspondingly larger values in the first row, as highlighted by the red boxes in Fig.1(b). The distinct branches within MBA exhibit varied attention patterns: some branches focus solely on a select few features, resulting in sparse attention maps (e.g., branch 0, where all heads exhibit sparsity); others engage in extensive feature interactions, leading to dense attention maps (e.g., branch 2, where most heads are dense); while certain branches achieve a balance between these two extremes (as illustrated by branch 1).

These diverse branches effectively enhance the diversity of attention outcomes, thereby augmenting interactions among heterogeneous features. Our experimental results further demonstrate that MBA achieves this diversity in a parameter-efficient manner. In addition, Fig.1(b) displays the branch weights for each branch, and the disparities among these weights underscore the distinctiveness and diversity of the different branches.

## 3.5 DECODER

The decoder features an inter-instance cross-attention block. Leveraging the encoder's instance-level representation extraction, we treat the instance set $\hat{Z} = \{\hat{z}_i\}$ as a sequence (full dataset or a batch, i.e. $|\hat{Z}| \leq N$), with each instance as a token (e.g., reshaping $[batchsize, d]$ to $[1, batchsize, d]$ in PyTorch). This reformulation enables direct use of encoder outputs as input for inter-instance attention. We integrate this mechanism with label information in a novel decoder block, combining cross-instance relational modeling and label-aware processing.

### 3.5.1 INTER-INSTANCE ATTENTION INCORPORATED WITH LABELS

We introduce IAIL (Inter-instance Attention Incorporated with Labels), a decoder component utilizing cross-attention (Fig.3(b)), which accepts multiple inputs. The $\ell$-th IAIL (denoted as $\text{IAIL}^\ell$) processes $\hat{Z}^{\ell-1} = \{\hat{z}_i^{\ell-1}\}$ to produce $\hat{Z}^\ell = \{\hat{z}_i^\ell\}$, with $\hat{Z}^0 = \{\hat{z}_i^0\}$ as initial input. To integrate label information, we employ a trainable Label Embedding Layer (LEL), projecting labels into an embedding space as:

$$\hat{y}_i = \text{LEL}(y_i), \hat{y}_i \in \mathbb{R}^d \tag{12}$$

where, for classification tasks, LEL acts as a lookup table, while for regression, it serves as a linear mapper. To align with feature embeddings, label embeddings are collectively denoted as $\hat{Y} = \{\hat{y}_i\}$.

During training and inference, IAIL's cross-attention mechanism receives distinct inputs, as follows:

- Training: for the $\ell$-th IAIL, the query comes from the prior IAIL's output, while the key uses the encoder's raw batch outputs, and the value incorporates the corresponding batch label information:

$$\hat{Z}_{batch}^{\ell-1} \to \text{Query}^\ell, \hat{Z}_{batch}^0 \to \text{Key}^\ell, \hat{Y}_{batch} \to \text{Value}^\ell \tag{13}$$

where the subscript "batch" refers to the group of instances in a specific batch.

- Inference: during inference, we utilize all instances from the training set as the input for both key and value:

$$\hat{z}_i^{\ell-1} \to \text{Query}^\ell, \hat{Z}_{train}^0 \to \text{Key}^\ell, \hat{Y}_{train} \to \text{Value}^\ell \tag{14}$$

where the subscript "i" denotes the $i$-th test instance and "train" the training dataset.

Following Gorishniy et al. (2024), we find that shared query/key projection weights occasionally improve model performance, prompting us to formalize this as a dataset-tunable hyperparameter. Meanwhile, we compute similarity via L2 distance instead of scaled dot-product. This eliminates the need for multi-head attention in IAIL.

Compared to Somepalli et al. (2021), IAIL achieves a more parameter-efficient inter-instance attention mechanism by leveraging the encoder's output, whereby each instance is compactly represented by a single [CLS] token. Incorporating label information enables the decoder to better learn the mapping from features to predictions. In Section 4.3, we demonstrate the efficacy of the IAIL block through ablation studies.

## 4 EXPERIMENTS

In this section, we compare the proposed MAYA with SOTA transformer-based methods on real-world classification/regression datasets to show its superiority and analyze its attributes via ablation studies.

## 4.1 Experimental Setting

**Datasets**   Our datasets are divided into two parts. For the first part (referred to as Part I in the following content), we select 14 datasets from previous studies Huang et al. (2020); Somepalli et al. (2021); Gorishniy et al. (2021), which have been widely utilized in the literature. For the second part (referred to as Part II in the following content), we employ a benchmark proposed by Grinsztajn et al. (2022), utilizing a broader range of datasets to test the superiority of MAYA. Detailed information and statistics about both parts of the datasets can be found in the Appendix B.1. For details regarding the partitioning and preprocessing of the datasets, please refer to Appendix B.2.

**Evaluation**   We primarily follow Gorishniy et al. (2021) for evaluation, using 15 random seeds per dataset and reporting average results along with standard deviations on test sets. Accuracy is used as metric for classification and Root Mean Squared Error (RMSE) for regression. We also report the average rank (lower is better) across datasets to assess overall performance.

**Comparison Methods**   The proposed MAYA will be benchmarked against several SOTA transformer-based methods, including AutoInt (Song et al., 2019), TabTransformer (Huang et al., 2020), SAINT (Somepalli et al., 2021), FT-Transformer (Gorishniy et al., 2021), ExcelFormer (Chen et al., 2024) and AMFormer (Cheng et al., 2024). All methods are based on their officially released implementations. Note that, for Datasets Part II, due to limitations in computational resources and time consumption, we do not compare with SAINT.

**Implementation Details**   For comparison methods, we cite results directly from papers if available; otherwise, we tune hyperparameters within their search spaces (see Appendix D). When multiple results exist, we cite the best. For our method, we use Optuna (Akiba et al., 2019) for 100 trials to find the optimal hyperparameters, applied uniformly across 15 random seeds. All hyperparameter tuning is based on training and validation sets.

## 4.2 Main Results

The comparison results between MAYA and other transformer-based methods on Datasets Part I and Part II are shown in Table1 and Table 2, respectively. Here, the first six datasets are selected from each of Part I and Part II for reporting. The results of the remaining datasets, along with their standard deviations, can be found in Appendix C. It is evident that MAYA significantly outperforms other methods in terms of the average rank, indicating that the overall performance of MAYA is currently the best among transformer-based models. Furthermore, as evidenced by the full results on Datasets Part II (Table 10 of the Appendix), MAYA demonstrates a particularly significant advantage on relatively larger datasets (where the number of training instances exceeds 9000).

Table 1: The results of all methods across the fisrt 5 datasets on Datasets Part I. All datasets are referred to by their abbreviations. For specific information about each dataset, please refer to Table 6. The scientific notation next to the dataset names indicates the scale of the results. The best result for each dataset is bolded. $\uparrow \sim$ accuracy (higher is better), $\downarrow \sim$ RMSE (lower is better).

| Dataset | AI | TT | SNT | FTT | EF | AMF | MAYA (ours) |
|---|---|---|---|---|---|---|---|
| AD $\uparrow$ | 0.8576 | 0.8509 | 0.8600 | 0.8588 | 0.8594 | 0.8594 | **0.8632** |
| BA $\uparrow$ | 0.9077 | 0.8998 | 0.9075 | 0.9095 | 0.9092 | 0.9082 | **0.9100** |
| BL $\uparrow$ | 0.7985 | 0.7775 | 0.8008 | 0.7995 | **0.8018** | 0.7990 | 0.8003 |
| CA $\downarrow$ | 0.5007 | 0.5936 | 0.4680 | 0.4564 | 0.4519 | 0.4626 | **0.4373** |
| DI$_{\times 10^3}$ $\downarrow$ | 0.5372 | 0.7345 | 0.5466 | 0.5334 | 0.5331 | 0.5384 | **0.5324** |
| rank | 4.5714 | 6.4286 | 4.0000 | 3.5714 | 3.6429 | 3.5000 | **2.2857** |

## 4.3 Ablation Studies

In this section, we undertake an exhaustive analysis of the design characteristics of MAYA, utilizing six diverse datasets:BA/BL/QS/SE for classification and CA/SH for regression.

Table 2: The results of all methods across the first 5 datasets on Datasets Part II. All datasets are referred to by their Dataset IDs. For specific information about each dataset, please refer to Table 7. The scientific notation next to the dataset names indicates the scale of the results. The best result for each dataset is bolded. $\uparrow \sim$ accuracy (higher is better), $\downarrow \sim$ RMSE (lower is better).

| Dataset ID | AI | TT | FTT | EF | AMF | MAYA (ours) |
|---|---|---|---|---|---|---|
| $01_{\times 10^{-3}} \downarrow$ | 0.1594 | 0.1890 | 0.1578 | 0.1583 | 0.1568 | **0.1556** |
| $02_{\times 10^2} \downarrow$ | 0.5215 | 1.1276 | **0.4301** | 0.4305 | 0.4328 | 0.4324 |
| $03 \uparrow$ | 0.7675 | 0.7363 | **0.8009** | 0.7991 | **0.8009** | 0.7959 |
| $04 \uparrow$ | 0.8580 | 0.8134 | **0.8620** | 0.8582 | 0.8590 | 0.8586 |
| $05 \downarrow$ | 0.1573 | 0.1732 | 0.1502 | 0.1481 | 0.1490 | **0.1456** |
| rank | 4.6818 | 5.9545 | 2.7727 | 2.9545 | 2.6818 | **1.9545** |

**The Properties of MBA**   We first investigate the impact of the MBA block's design details by degrading its attention mechanism to a single MHA (i.e., setting the number of parallel branches to 1). To ensure a fair comparison, we maintain the same total number of heads in MHA as in the MBA block of MAYA (i.e., num_heads$_{\text{MHA}}$ = num_heads$_{\text{per\_branch}} \times$ num_branch). Two configurations are tested:

- MHA$_{hidden\_size}$: aligns the hidden state dimension of MHA with that of the MBA block in MAYA;
- MHA$_{head\_dim}$: aligns the subspace dimension per head of MHA with that of the MBA block in MAYA.

All other architectural parameters, e.g., FFN intermediate scaling factor (i.e., intermediate_factor) and the number of blocks (i.e., num_layers) remain consistent with MAYA, while other hyperparameters are tuned as described in Section 4.1. Details are provided in Appendix E.1. Both configurations increase feature richness and diversity by expanding the number of heads. However,

- in MHA$_{hidden\_size}$, the subspace dimension per head is reduced, preserving overall feature richness but sacrificing per-subspace expressiveness;
- in MHA$_{head\_dim}$, the hidden state dimension grows, increasing FFN parameter size and computational overhead.

These comparisons highlight the MBA block's advantage in balancing feature complexity and parameter efficiency. Additionally, we perform a MHA$_{free}$ configuration by tuning all parameters except the branch count (set to 1) and conduct ablation studies on branch weights within the MBA block by removing them entirely and retuning hyperparameters.

The experimental results (Table 3) demonstrate that MAYA outperforms all ablation configurations of the MBA block, including the branch-weight-free variant, which also surpasses single-branch MHA settings. This suggests that MBA's branch-level feature averaging balances parameter efficiency and feature richness, while enhancing predictive capability akin to feature pooling. Branch weights further improve performance by promoting collaborative learning among branches. In contrast, MHA$_{hidden\_size}$ (which increases head count but reduces subspace dimensionality) yields the weakest results, indicating that such dimensionality reduction deteriorates predictive accuracy.

**The Properties of IAIL**   We conduct two experiments to investigate the impact of IAIL. In the first experiment, we remove the decoder and directly connect the Predictor to the encoder, denoted as "**w/o** decoder". In the second experiment, we retain the decoder but do not incorporate labels within the IAIL block, using the same input for both value and key, referred to as "IAIL **w/o** labels". All other hyperparameters are tuned according to the description in Section 4.1. The experimental results are presented in Table 4. It is evident that removing the decoder (i.e. removing the IAIL blocks), significantly impairs the model's predictive capability. Even without utilizing labels, integrating the decoder enhances the model's performance. Furthermore, incorporating labels leads to a further improvement in the model's predictive ability.

Table 3: The results of ablation studies regarding the MBA block. MHA denotes the use of only one attention branch, where the subscript "hidden_size" indicates that the hidden_size of this attention branch is consistent with that of MBA in MAYA. The subscript "head_dim" signifies that the head_dim of this attention branch aligns with that of MBA in MAYA. The subscript "free" indicates that all parameters are tuned. w/o $W_B$ represents the removal of the branch weight from MBA. The multiples $N\times$ following the results of $\text{MHA}_{hidden\_size}$ and $\text{MHA}_{head\_dim}$ indicate that the parameter count of the corresponding encoder is $N$ times that of MAYA. The computation details of $N$ are provided in Appendix E.2. The best result for each dataset is in bold. $\uparrow \sim$ accuracy (higher is better), $\downarrow \sim$ RMSE (lower is better).

| Dataset | $\text{MHA}_{hidden\_size}$ | $\text{MHA}_{head\_dim}$ | $\text{MHA}_{free}$ | w/o $W_B$ | MAYA |
|---|---|---|---|---|---|
| BA $\uparrow$ | 0.9084 (0.5$\times$) | **0.9101** (7.3$\times$) | 0.9092 | 0.9088 | 0.9100 |
| BL $\uparrow$ | 0.8000 (0.4$\times$) | 0.7995 (10.7$\times$) | 0.7969 | 0.8001 | **0.8003** |
| CA $\downarrow$ | 0.4479 (0.4$\times$) | 0.4412 (12.0$\times$) | 0.4405 | 0.4473 | **0.4373** |
| QS $\uparrow$ | **0.8578** (0.2$\times$) | 0.8559 (12.8$\times$) | 0.8288 | 0.8385 | 0.8515 |
| SE $\uparrow$ | 0.9275 (0.4$\times$) | 0.9278 (6.4$\times$) | 0.9287 | **0.9306** | 0.9299 |
| SH $\downarrow$ | 0.3161 (0.5$\times$) | 0.3166 (4.2$\times$) | 0.3143 | 0.3154 | **0.3126** |
| rank | 3.8333 | 3.1667 | 3.3333 | 3.0000 | **1.6667** |

Table 4: The results of ablation study regarding the IAIL block. The best result for each dataset is in bold. $\uparrow \sim$ accuracy (higher is better), $\downarrow \sim$ RMSE (lower is better).

| Dataset | BA $\uparrow$ | BL $\uparrow$ | CA $\downarrow$ | QS $\uparrow$ | SE $\uparrow$ | SH $\downarrow$ | rank |
|---|---|---|---|---|---|---|---|
| **w/o** decoder | 0.9088 | 0.7980 | 0.4411 | 0.8367 | 0.9248 | 0.3133 | 3.0000 |
| IAIL **w/o** labels | 0.9093 | 0.7983 | 0.4388 | 0.8392 | 0.9296 | **0.3117** | 1.8333 |
| MAYA | **0.9100** | **0.8003** | **0.4373** | **0.8515** | **0.9299** | 0.3126 | **1.1667** |

Table 5: The results of ablation study regarding the activation following the tokenizer. The best result for each dataset is in bold. $\uparrow \sim$ accuracy (higher is better), $\downarrow \sim$ RMSE (lower is better).

| Dataset | BA $\uparrow$ | BL $\uparrow$ | CA $\downarrow$ | QS $\uparrow$ | SE $\uparrow$ | SH $\downarrow$ |
|---|---|---|---|---|---|---|
| **w/o** activation in tokenizer | 0.9094 | 0.7997 | 0.4401 | 0.8346 | 0.9275 | 0.3129 |
| MAYA | **0.9100** | **0.8003** | **0.4373** | **0.8515** | **0.9299** | **0.3126** |

**The Influence of Activation after Tokenizer** We attempt to remove the activation function following the tokenizer, while maintaining all other configurations constant, and perform hyperparameter tuning according to the protocol outlined in Section 4.1. This configuration is subsequently compared against MAYA that retains the activation function, to ascertain its impact. The results, presented in Table 5, reveal that the incorporation of a ReLU activation function subsequent to tokenizer yields a marked enhancement in the model's predictive accuracy.

## 5 CONCLUSION

This paper introduces MAYA, a novel transformer-based encoder-decoder architecture for tabular data. Its dual-attention mechanism enhances intra- and inter-instance information fusion, while the Multi-Branch Attention (MBA) block leverages multiple attention branches to extract diverse features. Experiments on public datasets demonstrate MAYA's efficacy in tabular data processing. Ablation studies validate the effectiveness of MBA and IAIL blocks, while also demonstrating that incorporating activation functions following the tokenizer enhances the model's predictive performance.

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

## A ANALYSIS OF MBA

To obtain Fig.1(b) , we perform hyperparameter tuning on the shrutime (SH) dataset in accordance with the experimental configuration (Section 4.1 ), yielding the MAYA comprising 10 layers, with 3 parallel branches per layer and 4 attention heads per branch. The 5th layer is selected for visualization purposes. Furthermore, we utilize CatBoost (Prokhorenkova et al., 2018) to rank feature importance, identifying features 10, 2, 5, and 6 as the top four most important features. These features also frequently make substantial contributions to the [CLS] token, as exemplified by the red boxes in Fig.1(b) . Notably, the prominent contributions of these features typically manifest across distinct branches, thereby highlighting the necessity of employing multiple branches.

## B DETAILS ON DATASETS

### B.1 INFORMATION AND STATISTICS

In this section, we provide some detailed information and statistics about the datasets used in our experiments. Specifically, Table 6 presents the detailed information of the 14 datasets in Datasets Part I, which we summarized from previous works Huang et al. (2020); Somepalli et al. (2021); Gorishniy et al. (2021) and are widely used for evaluating the effectiveness of methods. Table 7 provides the detailed information of all the datasets in Datasets Part II, which originate from Grinsztajn et al. (2022), and we have selected 22 datasets from it.

Table 6: The details of Datasets Part I. "#Num" and "#Cat" denote the number of numerical and categorical features, respectively. Size denotes the number of all instances.

| Name | Abbr | Task | Size | #Num | #Cat | #Class | URL |
|------|------|------|------|------|------|--------|-----|
| Adult | AD | cls | 48842 | 6 | 8 | 2 | adult |
| Bank | BA | cls | 45211 | 7 | 9 | 2 | bank |
| Blastchar | BL | cls | 7032 | 3 | 16 | 2 | blastchar |
| California Housing | CA | reg | 20640 | 8 | 0 | - | california_housing |
| Diamonds | DI | reg | 53940 | 6 | 3 | - | diamonds |
| Helena | HE | cls | 65196 | 27 | 0 | 100 | helena |
| Income | IN | cls | 48842 | 6 | 8 | 2 | income |
| Jannis | JA | cls | 83733 | 54 | 0 | 4 | jannis |
| Otto Group Products | OT | cls | 61878 | 93 | 0 | 9 | otto_group_products |
| QSAR Bio | QS | cls | 1055 | 31 | 10 | 2 | qsar_bio |
| Seismic Bumps | SE | cls | 2584 | 14 | 4 | 2 | seismic_bumps |
| Shrutime | SH | reg | 10000 | 4 | 6 | - | shrutime |
| Spambase | SP | cls | 4601 | 57 | 0 | 2 | spambase |
| Volume | VO | reg | 50993 | 53 | 0 | - | volume |

Table 7: The details of Datasets Part II. "#Num" and "#Cat" denote the number of numerical and categorical features, respectively. #Training means the number of instances in the training set.

| Name | Dataset ID | Task | #Training | #Num | #Cat | #Class |
|------|-----------|------|-----------|------|------|--------|
| Ailerons | 01 | reg | 9625 | 33 | 0 | - |
| BikeSharingDemand | 02 | reg | 10000 | 6 | 5 | - |
| KDDCup09upselling | 03 | cls | 3589 | 34 | 15 | 2 |
| MagicTelescope | 04 | cls | 9363 | 10 | 0 | 2 |
| MiamiHousing2016 | 05 | reg | 9752 | 13 | 0 | - |
| OnlineNewsPopularity | 06 | reg | 9999 | 45 | 15 | - |
| blackfriday | 07 | reg | 50000 | 4 | 5 | - |
| california | 08 | reg | 10000 | 8 | 0 | - |
| compass | 09 | cls | 10000 | 8 | 9 | 2 |
| cpuact | 10 | reg | 5734 | 21 | 0 | - |
| credit | 11 | cls | 10000 | 10 | 0 | 2 |
| electricity | 12 | cls | 10000 | 7 | 1 | 2 |
| elevators | 13 | reg | 10000 | 16 | 0 | - |
| fifa | 14 | reg | 10000 | 5 | 0 | - |
| housesales | 15 | reg | 10000 | 15 | 2 | - |
| houses | 16 | reg | 10000 | 8 | 0 | - |
| kddipumsla97-small | 17 | cls | 3631 | 20 | 0 | 2 |
| particulate-matter-ukair-2017 | 18 | reg | 50000 | 3 | 3 | - |
| phoneme | 19 | cls | 2220 | 5 | 0 | 2 |
| pol | 20 | reg | 10000 | 26 | 0 | - |
| rl | 21 | cls | 3749 | 5 | 7 | 2 |
| winequality | 22 | reg | 4547 | 11 | 0 | - |

### B.2 PREPROCESSING

For Datasets Part I, we generally adhere to the preprocessing methodology outlined by Gorishniy et al. (2021), partitioning the dataset into training, validation, and test sets in ratios of 64%/16%/20%, respectively. Normalization is performed using quantile transformation from Scikit-learn library (Pedregosa et al., 2011) on the datasets.

Regarding Datasets Part II, we follow the preprocessing procedures specified in Grinsztajn et al. (2022); Gorishniy et al. (2024). We utilize the identical partitioning ratios as those in Grinsztajn et al. (2022) and truncate the training set (to 10,000 or 50,000 instances). In accordance with Gorishniy et al. (2024), if a dataset comprises $n$ splits (corresponding to the $n$-fold cross-validation mentioned in Grinsztajn et al. (2022)), each split is treated as an independent dataset for tuning and evaluation. Subsequently, the results across these $n$ splits are averaged to derive the final outcome for the dataset, effectively averaging over $n \times 15$ seeds. The normalization method employed is identical to that used in Datasets Part I.

## C MORE RESULTS

In this section, we present all the experimental results, including the average results and standard deviations of each method on Datasets Part I and Part II, as well as the average rankings. Specifically, the average results for Datasets Part I are shown in Table 8, with the corresponding standard deviations in Table 9. The average results for Datasets Part II are presented in Table 10, and their standard deviations are in Table 11. The experimental results underscore the superiority of MAYA, particularly evident in the Datasets Part II findings (Table 10), where its advantages become pronounced on relatively larger-scale datasets (where the number of training instances exceeds 9000).

Table 8: The results of all methods across 14 datasets on Datasets Part I. The upward arrow (↑) indicates the accuracy for classification tasks (higher is better), while the downward arrow (↓) represents the RMSE for regression tasks (lower is better). The average rank results also follow the principle that lower is better. The scientific notation next to the dataset names indicates the scale of the results. The best result for each dataset is bolded.

| Dataset | AI | TT | SNT | FTT | EF | AMF | MAYA (ours) |
|---|---|---|---|---|---|---|---|
| AD ↑ | 0.8576 | 0.8509 | 0.8600 | 0.8588 | 0.8594 | 0.8594 | **0.8632** |
| BA ↑ | 0.9077 | 0.8998 | 0.9075 | 0.9095 | 0.9092 | 0.9082 | **0.9100** |
| BL ↑ | 0.7985 | 0.7775 | 0.8008 | 0.7995 | **0.8018** | 0.7990 | 0.8003 |
| CA ↓ | 0.5007 | 0.5936 | 0.4680 | 0.4564 | 0.4519 | 0.4626 | **0.4373** |
| DI$_{\times 10^3}$ ↓ | 0.5372 | 0.7345 | 0.5466 | 0.5334 | 0.5331 | 0.5384 | **0.5324** |
| HE ↑ | 0.3811 | 0.3589 | 0.3856 | **0.3890** | 0.3802 | 0.3882 | 0.3831 |
| IN ↑ | 0.8605 | 0.8535 | 0.8661 | 0.8633 | 0.8665 | 0.8625 | **0.8681** |
| JA ↑ | 0.7178 | 0.7084 | 0.7111 | **0.7306** | 0.7262 | 0.7281 | 0.7207 |
| OT ↑ | 0.8084 | 0.8019 | **0.8120** | 0.8082 | 0.8035 | 0.8074 | 0.7984 |
| QS ↑ | 0.8357 | 0.8461 | 0.8452 | 0.8330 | 0.8389 | **0.8537** | 0.8515 |
| SE ↑ | **0.9309** | 0.9243 | 0.9259 | 0.9275 | 0.9282 | 0.9253 | 0.9299 |
| SH ↓ | 0.3255 | 0.3464 | 0.3131 | 0.3222 | 0.3197 | 0.3149 | **0.3126** |
| SP ↑ | 0.9362 | 0.9393 | 0.9294 | 0.9424 | 0.9346 | **0.9435** | 0.9425 |
| VO$_{\times 10^2}$ ↓ | **0.3987** | 0.4940 | 0.4171 | 0.4210 | 0.4253 | 0.4097 | 0.4037 |
| rank | 4.5714 | 6.4286 | 4.0000 | 3.5714 | 3.6429 | 3.5000 | **2.2857** |

## D HYPERPARAMETER TUNING SPACE

Here we provide hyperparameter tuning spaces used for Optuna tuning for all methods in Section 4.2.

Table 9: The standard deviations of all methods across 14 datasets on Datasets Part I.

| Dataset | AI | TT | SNT | FTT | EF | AMF | MAYA (ours) |
|---|---|---|---|---|---|---|---|
| AD ↑ | 0.0011 | 0.0016 | 0.0020 | 0.0014 | 0.0014 | 0.0012 | 0.0017 |
| BA ↑ | 0.0016 | 0.0016 | 0.0012 | 0.0012 | 0.0012 | 0.0017 | 0.0012 |
| BL ↑ | 0.0031 | 0.0037 | 0.0025 | 0.0017 | 0.0027 | 0.0042 | 0.0021 |
| CA ↓ | 0.0034 | 0.0022 | 0.0050 | 0.0036 | 0.0049 | 0.0051 | 0.0034 |
| DI ↓ | 5.8841 | 12.317 | 2.7438 | 4.5085 | 6.8204 | 4.0224 | 6.3196 |
| HE ↑ | 0.0015 | 0.0020 | 0.0025 | 0.0014 | 0.0023 | 0.0019 | 0.0021 |
| IN ↑ | 0.0009 | 0.0013 | 0.0022 | 0.0026 | 0.0015 | 0.0017 | 0.0015 |
| JA ↑ | 0.0018 | 0.0020 | 0.0015 | 0.0020 | 0.0021 | 0.0020 | 0.0021 |
| OT ↑ | 0.0020 | 0.0021 | 0.0020 | 0.0028 | 0.0027 | 0.0029 | 0.0049 |
| QS ↑ | 0.0119 | 0.0642 | 0.0126 | 0.0159 | 0.0170 | 0.0147 | 0.0091 |
| SE ↑ | 0.0031 | 0.0023 | 0.0027 | 0.0067 | 0.0067 | 0.0028 | 0.0024 |
| SH ↓ | 0.0028 | 0.0027 | 0.0014 | 0.0044 | 0.0014 | 0.0019 | 0.0020 |
| SP ↑ | 0.0027 | 0.0030 | 0.0029 | 0.0036 | 0.0031 | 0.0037 | 0.0039 |
| VO ↓ | 0.7686 | 3.2592 | 1.6812 | 1.3830 | 0.9772 | 1.4935 | 1.1377 |

Table 10: The results of all methods across 22 datasets on Datasets Part II. All datasets are referred to by their Dataset IDs. For specific information about each dataset, please refer to Table 7. The upward arrow (↑) indicates the accuracy for classification tasks (higher is better), while the downward arrow (↓) represents the RMSE for regression tasks (lower is better). The average rank results also follow the principle that lower is better. The scientific notation next to the dataset names indicates the scale of the results. The best result for each dataset is bolded.

| Dataset ID | AI | TT | FTT | EF | AMF | MAYA (ours) |
|---|---|---|---|---|---|---|
| $01_{\times 10^{-3}}$ ↓ | 0.1594 | 0.1890 | 0.1578 | 0.1583 | 0.1568 | **0.1556** |
| $02_{\times 10^{2}}$ ↓ | 0.5215 | 1.1276 | **0.4301** | 0.4305 | 0.4328 | 0.4324 |
| 03 ↑ | 0.7675 | 0.7363 | **0.8009** | 0.7991 | **0.8009** | 0.7959 |
| 04 ↑ | 0.8580 | 0.8134 | **0.8620** | 0.8582 | 0.8590 | 0.8586 |
| 05 ↓ | 0.1573 | 0.1732 | 0.1502 | 0.1481 | 0.1490 | **0.1456** |
| 06 ↓ | 0.8653 | 0.8751 | 0.8639 | **0.8623** | 0.8646 | 0.8666 |
| 07 ↓ | 0.3698 | 0.4732 | 0.3673 | 0.3661 | 0.3666 | **0.3650** |
| 08 ↓ | 0.1441 | 0.1837 | 0.1381 | 0.1374 | 0.1386 | **0.1341** |
| 09 ↑ | 0.7618 | 0.7499 | 0.7729 | 0.7478 | 0.7627 | **0.7912** |
| $10_{\times 10}$ ↓ | 0.2332 | 0.5714 | 0.2219 | 0.2261 | **0.2186** | 0.2226 |
| 11 ↑ | 0.7745 | 0.7595 | 0.7743 | 0.7747 | **0.7754** | 0.7730 |
| 12 ↑ | 0.8203 | 0.8000 | 0.8339 | 0.8339 | 0.8311 | **0.8434** |
| $13_{\times 10^{-2}}$ ↓ | 0.1861 | 0.2802 | 0.1828 | 0.1854 | 0.1830 | **0.1795** |
| 14 ↓ | 0.7973 | 1.1280 | 0.7922 | 0.7923 | 0.7905 | **0.7873** |
| 15 ↓ | 0.1754 | 0.1999 | 0.1668 | 0.1677 | 0.1670 | **0.1653** |
| 16 ↓ | 0.2322 | 0.2926 | **0.2247** | 0.2281 | 0.2290 | 0.2269 |
| 17 ↑ | 0.8800 | 0.8771 | 0.8823 | 0.8827 | **0.8841** | 0.8832 |
| 18 ↓ | 0.3775 | 0.5275 | 0.3750 | 0.3696 | 0.3733 | **0.3658** |
| 19 ↑ | 0.8587 | 0.8207 | 0.8667 | 0.8692 | **0.8708** | 0.8675 |
| $20_{\times 10}$ ↓ | 0.3797 | 1.9690 | 0.2580 | 0.2824 | 0.2667 | **0.2547** |
| 21 ↑ | 0.6832 | 0.6206 | 0.7234 | 0.7358 | 0.7288 | **0.7662** |
| 22 ↓ | 0.6728 | 0.6931 | 0.6802 | 0.6794 | 0.6802 | **0.6706** |
| rank | 4.6818 | 5.9545 | 2.7727 | 2.9545 | 2.6818 | **1.9545** |

Table 11: The standard deviations of all methods across 22 datasets on Datasets Part II.

| Dataset ID | AI | TT | FTT | EF | AMF | MAYA (ours) |
|---|---|---|---|---|---|---|
| 01 ↓ | 0.0000 | 0.0000 | 0.0000 | 0.0000 | 0.0000 | 0.0000 |
| 02 ↓ | 10.370 | 0.9339 | 0.4644 | 0.6977 | 0.7884 | 0.8374 |
| 03 ↑ | 0.0096 | 0.0115 | 0.0096 | 0.0137 | 0.0090 | 0.0112 |
| 04 ↑ | 0.0056 | 0.0067 | 0.0035 | 0.0047 | 0.0073 | 0.0053 |
| 05 ↓ | 0.0041 | 0.0062 | 0.0032 | 0.0030 | 0.0028 | 0.0015 |
| 06 ↓ | 0.0009 | 0.0014 | 0.0008 | 0.0015 | 0.0009 | 0.0005 |
| 07 ↓ | 0.0008 | 0.0197 | 0.0007 | 0.0008 | 0.0008 | 0.0006 |
| 08 ↓ | 0.0009 | 0.0009 | 0.0011 | 0.0015 | 0.0012 | 0.0012 |
| 09 ↑ | 0.0065 | 0.0072 | 0.0071 | 0.0076 | 0.0123 | 0.0257 |
| 10 ↓ | 0.0535 | 0.3803 | 0.0694 | 0.0694 | 0.0531 | 0.0551 |
| 11 ↑ | 0.0043 | 0.0044 | 0.0044 | 0.0045 | 0.0042 | 0.0031 |
| 12 ↑ | 0.0020 | 0.0015 | 0.0027 | 0.0030 | 0.0023 | 0.0046 |
| 13 ↓ | 0.0000 | 0.0000 | 0.0000 | 0.0000 | 0.0000 | 0.0000 |
| 14 ↓ | 0.7973 | 1.1280 | 0.7922 | 0.7923 | 0.0120 | 0.7873 |
| 15 ↓ | 0.0010 | 0.0008 | 0.0007 | 0.0008 | 0.0007 | 0.0005 |
| 16 ↓ | 0.0024 | 0.0021 | 0.0008 | 0.0018 | 0.0044 | 0.0018 |
| 17 ↑ | 0.0044 | 0.0051 | 0.0062 | 0.0049 | 0.0060 | 0.0050 |
| 18 ↓ | 0.0019 | 0.0003 | 0.0014 | 0.0011 | 0.0019 | 0.0009 |
| 19 ↑ | 0.0121 | 0.0094 | 0.0090 | 0.0110 | 0.0116 | 0.0128 |
| 20 ↓ | 0.1161 | 0.1567 | 0.1317 | 0.1240 | 0.1253 | 0.1241 |
| 21 ↑ | 0.0127 | 0.0159 | 0.0148 | 0.0158 | 0.0145 | 0.0106 |
| 22 ↓ | 0.0153 | 0.0166 | 0.0155 | 0.0055 | 0.0149 | 0.0250 |

## D.1 MAYA

The hyperparameter tuning space for MAYA is presented in Table 12.

## D.2 AUTOINT

The hyperparameter tuning space for AutoInt is presented in Table 13.

## D.3 TABTRANSFORMER

The hyperparameter tuning space for TabTransformer is presented in Table 14.

## D.4 SAINT

The hyperparameter tuning space for SAINT is presented in Table 15.

## D.5 FT-TRANSFORMER

The hyperparameter tuning space for FT-Transformer is presented in Table 16.

## D.6 EXCELFORMER

The hyperparameter tuning space for ExcelFormer is presented in Table 17.

## D.7 AMFORMER

The hyperparameter tuning space for AMFormer is presented in Table 18.

Table 12: Hyperparameter tuning space for MAYA.

|  | Parameter | Distribution |
|---|---|---|
| | num_layers | $\text{UniformInt}[1, 16]$ |
| | hidden_size | $\text{UniformInt}[64, 256]$ |
| | num_heads | $\text{Categorical}[1, 4, 8, 32]$ |
| | num_branch | $\text{UniformInt}[2, 8]$ |
| encoder | intermediate_factor | $\text{Categorical}[0.8, 1, 1.3, 2]$ |
| | dropout | $\text{Uniform}[0, 0.3]$ |
| | add_act | $\text{Categorical}[\text{true}, \text{false}]$ |
| | if_bias | $\text{Categorical}[\text{true}, \text{false}]$ |
| | act_type | $\text{Categorical}[\text{relu}, \text{prelu}]$ |
| | num_decoder_layers | $\text{UniformInt}[1, 16]$ |
| | decoder_intermediate_factor | $\text{Categorical}[0.8, 1, 1.3, 2]$ |
| decoder | decoder_dropout | $\text{Uniform}[0, 0.3]$ |
| | decoder_if_bias | $\text{Categorical}[\text{true}, \text{false}]$ |
| | decoder_act_type | $\text{Categorical}[\text{relu}, \text{prelu}]$ |
| | qk_shared_weights | $\text{Categorical}[\text{true}, \text{false}]$ |
| | batch_size | $\text{Categorical}[1024, 2048, 4096]$ |
| others | learning_rate | $\text{LogUniform}[1e\text{-}5, 5e\text{-}3]$ |
| | weight_decay | $\text{Uniform}[0, 0.3]$ |

Table 13: Hyperparameter tuning space for AutoInt.

| Parameter | Distribution |
|---|---|
| n_layers | $\text{UniformInt}[1, 6]$ |
| d_token | $\text{UniformInt}[8, 64]$ |
| residual_dropout | $\{0, \text{Uniform}[0, 0.2]\}$ |
| attention_dropout | $\{0, \text{Uniform}[0, 0.5]\}$ |
| learning_rate | $\text{LogUniform}[1e\text{-}5, 1e\text{-}3]$ |
| weight_decay | $\text{LogUniform}[1e\text{-}6, 1e\text{-}3]$ |

Table 14: Hyperparameter tuning space for TabTransformer.

| Parameter | Distribution |
|---|---|
| dim | $\text{Categorical}[32, 64, 128, 256]$ |
| depth | $\text{Categorical}[1, 2, 3, 6, 12]$ |
| heads | $\text{Categorical}[2, 4, 8]$ |
| attn_dropout | $\text{Uniform}[0, 0.5]$ |
| ffn_dropout | $\text{Uniform}[0, 0.5]$ |
| learning_rate | $\text{LogUniform}[1e\text{-}6, 1e\text{-}3]$ |
| weight_decay | $\text{LogUniform}[1e\text{-}6, 1e\text{-}1]$ |

Table 15: Hyperparameter tuning space for SAINT.

| Parameter | Distribution |
|---|---|
| dim | Categorical$[16, 32, 64]$ |
| depth | Categorical$[4, 6]$ |
| heads | Categorical$[4, 8]$ |
| attn_dropout | Uniform$[0, 0.5]$ |
| ffn_dropout | Uniform$[0, 0.5]$ |
| attn_type | $\{$colrow, Categorical$[$colrow, row, col$]\}$ |
| learning_rate | LogUniform$[3e\text{-}5, 1e\text{-}3]$ |
| weight_decay | $\{0, $LogUniform$[1e\text{-}6, 1e\text{-}4]\}$ |

Table 16: Hyperparameter tuning space for FT-Transformer.

| Parameter | Distribution |
|---|---|
| n_layers | UniformInt$[1, 4]$ |
| d_token | UniformInt$[64, 512]$ |
| residual_dropout | $\{0, $Uniform$[0, 0.2]\}$ |
| attention_dropout | Uniform$[0, 0.5]$ |
| ffn_dropout | Uniform$[0, 0.5]$ |
| ffn_factor | Uniform$[2/3, 8/3]$ |
| learning_rate | LogUniform$[1e\text{-}5, 1e\text{-}3]$ |
| weight_decay | LogUniform$[1e\text{-}6, 1e\text{-}3]$ |

Table 17: Hyperparameter tuning space for ExcelFormer.

| Parameter | Distribution |
|---|---|
| n_layers | UniformInt$[2, 5]$ |
| d_token | Categorical$[64, 128, 256]$ |
| n_heads | Categorical$[4, 8, 16, 32]$ |
| residual_dropout | Uniform$[0, 0.5]$ |
| mix_type | Categorical$[$none, feat_mix, hidden_mix$]$ |
| learning_rate | LogUniform$[3e\text{-}5, 1e\text{-}3]$ |
| weight_decay | $\{0, $LogUniform$[1e\text{-}6, 1e\text{-}3]\}$ |

Table 18: Hyperparameter tuning space for AMFormer.

| Parameter | Distribution |
|---|---|
| dim | Categorical$[8, 16, 32, 64, 128]$ |
| depth | Categorical$[1, 4]$ |
| attn_dropout | Uniform$[0, 0.5]$ |
| ffn_dropout | Uniform$[0, 0.5]$ |
| num_special_tokens | UniformInt$[1, 4]$ |
| learning_rate | LogUniform$[1e\text{-}5, 1e\text{-}3]$ |
| weight_decay | LogUniform$[1e\text{-}6, 1e\text{-}3]$ |

# E ABLATION STUDIES

## E.1 IMPLEMENTATION DETAILS

In this section, we provide implementation details of the ablation studies on the properties of MBA block. The first two experimental configurations (i.e., $\text{MHA}_{hidden\_size}$ and $\text{MHA}_{head\_dim}$) ensure that the number of heads in MHA matches that in MBA of MAYA, which is formulated as:

$$\text{num\_head}_{\text{MHA}} = \text{num\_heads}_{per\_branch} \times \text{num\_branch} \tag{15}$$

For the first configuration, $\text{MHA}_{hidden\_size}$, we ensure that the hidden_size is consistent with MBA. The hidden_size for MHA should be:

$$\text{hidden\_size}_{\text{MHA}} = \text{head\_dim}_{\text{MHA}} \times \text{num\_head}_{\text{MHA}} \tag{16}$$

However, there may be cases where the hidden_size cannot be evenly divided by the number of parallel attention branches (i.e., num_branch), resulting in a non-integer value for $\text{head\_dim}_{\text{MHA}}$ in Eq.16. In such cases, we perform a ceiling operation on $\text{head\_dim}_{\text{MHA}}$, which may lead to $\text{hidden\_size}_{\text{MHA}}$ being slightly larger than $\text{hidden\_size}_{\text{MBA}}$. Please refer to Table 19 for the specific hidden sizes corresponding to each dataset.

For the second configuration, $\text{MHA}_{head\_dim}$, we ensure that the subspace dimension for each head, i.e. $\text{head\_dim}_{\text{MHA}}$, remains identical with MBA, whereby the hidden_size for MHA is still calculated using Eq.16. For the specific hidden sizes corresponding to each dataset, please refer to Table 20.

Table 19: Details for $\text{MHA}_{hidden\_size}$. Notations: $\#_{branch} \sim \text{num\_branch}$, $\#_{heads} \sim \text{num\_heads}$, $D_{attn} \sim \text{hidden\_size}$, $D_{head} \sim \text{head\_dim}$.

| | MBA in MAYA | | | $\text{MHA}_{hidden\_size}$ | | |
| --- | --- | --- | --- | --- | --- | --- |
| | $\#_{branch}$ | $\#_{heads}$ | $D_{attn}$ | $\#_{heads}$ | $D_{head}$ | $D_{attn}$ |
| BA | 4 | 4 | 192 | $4 \times 4 = 16$ | $192/16 = 12$ | $16 \times 12 = 192$ |
| BL | 6 | 8 | 128 | $6 \times 8 = 48$ | $\lceil 128/48 \rceil = 3$ | $48 \times 3 = 144$ |
| CA | 6 | 1 | 128 | $6 \times 1 = 6$ | $\lceil 128/6 \rceil = 22$ | $6 \times 22 = 132$ |
| QS | 8 | 1 | 160 | $8 \times 1 = 8$ | $160/8 = 20$ | $8 \times 20 = 160$ |
| SE | 4 | 1 | 224 | $4 \times 1 = 4$ | $224/4 = 56$ | $4 \times 56 = 224$ |
| SH | 3 | 4 | 224 | $3 \times 4 = 12$ | $\lceil 224/12 \rceil = 19$ | $12 \times 19 = 228$ |

Table 20: Details for $\text{MHA}_{head\_dim}$. Notations: $\#_{branch} \sim \text{num\_branch}$, $\#_{heads} \sim \text{num\_heads}$, $D_{attn} \sim \text{hidden\_size}$, $D_{head} \sim \text{head\_dim}$.

| | MBA in MAYA | | | | $\text{MHA}_{head\_dim}$ | | |
| --- | --- | --- | --- | --- | --- | --- | --- |
| | $\#_{branch}$ | $\#_{heads}$ | $D_{attn}$ | $D_{head}$ | $\#_{heads}$ | $D_{head}$ | $D_{attn}$ |
| BA | 4 | 4 | 192 | $192/4 = 48$ | $4 \times 4 = 16$ | 48 | $16 \times 48 = 768$ |
| BL | 6 | 8 | 128 | $128/8 = 16$ | $6 \times 8 = 48$ | 16 | $48 \times 16 = 768$ |
| CA | 6 | 1 | 128 | $128/1 = 128$ | $6 \times 1 = 6$ | 128 | $6 \times 128 = 768$ |
| QS | 8 | 1 | 160 | $160/1 = 160$ | $8 \times 1 = 8$ | 160 | $8 \times 160 = 1280$ |
| SE | 4 | 1 | 224 | $224/1 = 224$ | $4 \times 1 = 4$ | 224 | $4 \times 224 = 896$ |
| SH | 3 | 4 | 224 | $224/4 = 56$ | $3 \times 4 = 12$ | 56 | $12 \times 56 = 672$ |

## E.2 COMPUTATION OF PARAMETER COUNTS IN TABEL 3 OF THE MAIN TEXT

In Table 3 of the main text, we present a comparison of the parameter counts in the encoder of $\text{MHA}_{hidden\_size}$ and $\text{MHA}_{head\_dim}$ with that of MAYA. Here, we provide the specific method for the calculation.

In the MBA block, each attention branch constitutes an MHA. The MHA is comprised of four linear layers, namely query, key, value, and output projection. The weights of each linear layer form a

square matrix of dimensions hidden_size $\times$ hidden_size. Therefore, the number of parameters in the attention mechanism can be calculated as follows:

$$P_{attn} = \text{num\_branch} \times (4 \times \text{hidden\_size}^2) \tag{17}$$

In the FFN, we employ ReGLU (Shazeer, 2020), which differs from the original transformer architecture by incorporating three linear layers. Analogous to the naming convention used in LLaMA (Touvron et al., 2023), we refer to these layers as up, gate, and down projections. Due to the presence of the intermediate_factor , the matrix weights of each linear layer are of dimension intermediate_factor $\times$ hidden_size $\times$ hidden_size. Consequently, the number of parameters in the FFN can be calculated as follows:

$$P_{ffn} = 3 \times \text{intermediate\_factor} \times \text{hidden\_size}^2 \tag{18}$$

Combining Eq.17 and Eq.18, the number of parameters in an encoder with num_layers of blocks is calculated as follows:

$$P = \text{num\_layers} \times (P_{attn} + P_{ffn}) \tag{19}$$
$$= \text{num\_layers} \times \text{hidden\_size}^2 \tag{20}$$
$$\times (4 \times \text{num\_branch} + 3 \times \text{intermediate\_factor}) \tag{21}$$

According to Eq.19, when calculating the number of parameters for the encoders of $\text{MHA}_{hidden\_size}$, $\text{MHA}_{head\_dim}$, and MAYA, one simply substitutes the corresponding variables into the equation. Note that the num_layers and intermediate_factor for both MHA and MAYA are always kept consistent. For instance, for the BA dataset (intermediate_factor= 2, num_layers= 15), the number of parameters for the encoders of $\text{MHA}_{hidden\_size}$, $\text{MHA}_{head\_dim}$, and MAYA are as follows, respectively:

$$P_{\text{MHA}_{hidden\_size}} = 15 \times 192^2 \times (4 \times 1 + 3 \times 2) \tag{22}$$
$$P_{\text{MHA}_{head\_dim}} = 15 \times 768^2 \times (4 \times 1 + 3 \times 2) \tag{23}$$
$$P_{\text{MAYA}} = 15 \times 192^2 \times (4 \times 4 + 3 \times 2) \tag{24}$$

Therefore, the multiples presented in Table 3 of the main text are

$$P_{\text{MHA}_{hidden\_size}}/P_{\text{MAYA}} \approx 0.5 \tag{25}$$
$$P_{\text{MHA}_{head\_dim}}/P_{\text{MAYA}} \approx 7.3 \tag{26}$$

## F  LIMITATIONS

MAYA presents a novel baseline framework that adapts transformer architecture for tabular data, but several limitations remain. Firstly, due to limitations in computational resources and time constraints, we do not perform experiments on large-scale datasets (where instance size is on the order of $10^5$). In the future, we will investigate the performance of the proposed method on a larger benchmark. Secondly, a memory bottleneck exists due to the full pairwise L2 computation during the model inference stage. Although grouping every G instances of the training set and calculating them separately can alleviate this problem (see Fig.4), we hope to avoid calculating the full pairwise L2 with the entire training dataset during the inference stage in future work.

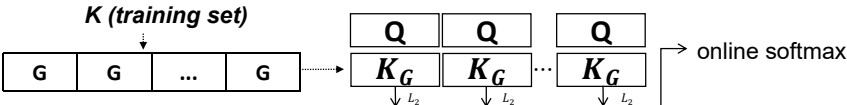

Figure 4: Calculate K in groups

