# OpenReview forum: "Multi-branch of Attention Yields Accurate Results for Tabular Data"
_ICLR.cc/2026/Conference — Submitted to ICLR 2026_

### Official Review · Reviewer_DgCZ · 2025-10-20

**Soundness:** 3
**Presentation:** 3
**Contribution:** 2
**Rating:** 2
**Confidence:** 4

**Summary:**

The paper proposes a new attention mechanism tailored for tabular data. The authors first identify two drawbacks of standard multi-head attention (MHA) for heterogeneous features: (1) that increasing the number of heads ($H$) to capture feature diversity unnecessarily increases FFN parameters, and (2) that the final projection layer mixes subspaces, reducing diversity.

As a solution, they propose Multi-Branch Attention (MBA), which computes several MHA blocks in parallel and combines their outputs using a weighted average. These weights are learned by a predictor network based on the branch outputs.

Second, the paper introduces an inter-instance attention with labels (IAIL) prediction scheme. IAIL uses a decoder architecture where queries are the test instance features, while keys and values are the training batch features and label embeddings, respectively. A final MLP head forms the prediction.

**Strengths:**

1. **Novel architecture.** The proposed architecture appears novel. While the proposed MBA scheme is a bit confusing, I find the IAIL interesting. It reminds me of the ModernNCA (Ye 2025) approach but adapted to the transformer architecture.
2. **Clarity.** The paper is relatively easy to follow and is well-written.
3. **Results.** The authors provide an ablation section, which highlights the importance of MBA and IAIL. The authors also provide a brief comparison of their method with other tabular Transformer architectures on a selected set of open-source datasets.

**Weaknesses:**

1. **MBA formulation, motivation and ablation.** I find the design of the MBA block confusing. First, I want to establish that in the standard Transformer implementation, the number of heads is the main driver of the diversity. The authors argue that this is inadequate for two reasons:
>  two drawbacks: (1) concatenating MHA head subspaces enlarges attention output hidden states, thus increasing the number of parameters in FFN and computational cost; (2) output projection of concatenated features causes mixing, reducing subspace diversity.

For reason (1), I feel this is incorrect usage of terminology: concatenating heads does not enlarge hidden space, it just increases the projection matrix, $W_O$, dimensions. For reason (2), I find it unsubstantiated, do you have reference works or anything to back this story?

In general, I think the design choice of MBA is not canonical, i.e., there are multiple ways to achieve this, which makes the work less interesting. For example, from the authors' own ablation Table 3, it seems it's the introduction of the weights $W_B$ that is crucial for the performance. However, these weights could be similarly computed for the heads, and then the head outputs could be similarly combined.

2. **The IAIL block.** I think the formulation is very close to the one proposed in ModernNCA (Ye 2024). Therefore, I would want to see a close comparison and the differences to be highlighted.

3. **Lacking experimental results.** Transformers are not SOTA in tabular, except when they are used in ICL context such as in TabPFN, TabICL, and Limix. Why pursue this direction? I would want to see a comparison of the proposed MAYA against the SOTA tabular models. For example, for California, I've seen models achieving below 0.4 in RMSE.

**Questions:**

Given that the model does not demonstrate state-of-the-art performance and the paper lacks theoretical analysis, what is the primary intended contribution of this work? Are the methodological components (MBA, IAIL) intended to provide generalizable insights? Please clarify how this work should be positioned within the broader literature on tabular deep learning.

---

### Official Review · Reviewer_aEuc · 2025-10-29

**Soundness:** 2
**Presentation:** 2
**Contribution:** 1
**Rating:** 2
**Confidence:** 3

**Summary:**

The paper proposes a novel, transformer-based architecture for classification and regression on tabular data using an encoder/decoder framework called MAYA. The work introduces two new layer types: multi-branch attention in the encoder and Inter--instance Attention Incorporating with Labels (IAIL) in the decoder.

The paper compares classification performance on 5 regression datasets from the literature, and the Grinsztajn benchmark suite, showing the superiority of the proposed method over some previously published transformer based architectures.

**Strengths:**

- The paper uses a well-established benchmark suite, the Grinsztajn benchmark.
- The paper addresses heterogeneous feature distributions, a common source of low performance for neural architectures.
- The paper does extensive hyper-parameter tuning for all methods, and provides the hyper-parameter spaces that were used.
- The paper provides ablation studies for some of the architectural choices.

**Weaknesses:**

- The paper does not compare against any state-of-the-art algorithms for regression and classification for tabular data. Limiting the comparison to "transformer-based architectures" is not a meaningful constraint to me. Why would one be interested in the best transformer-based architecture, and not in the best architecture overall? The authors also exclude state-of-the-art pre-trained transformer methods like TabPFNV2, TabDBT and TabICL. A good overview of state-of-the-art methods for tabular classification and regression can be found in the TabArena benchmark: https://huggingface.co/spaces/TabArena/leaderboard  https://huggingface.co/spaces/TabArena/leaderboard

- The only algorithm considered in the paper that had better than baseline performance on the TabZilla benchmark https://arxiv.org/pdf/2305.02997 is SAINT, and this method was not evaluated on the full benchmark.

- TabICL in particular has addressed feature heterogeneity through pre-trained column embeddings. The benefit over this approach should be clearly demonstrated in the paper.

- No comparison of training and inference times are provided in the paper.

The main question about any new architecture is "does it work". It's unclear that the proposed architecture is superior to the relatively simple TabM or RealMLP, for example.

**Questions:**

- Can you either submit results to tabarena, for a fair comparison to state-of-the-art methods, or compare against TabPFNV2, TabDBT, RealMLP and TabM on your benchmark?

- Can you please provide training and inference time comparisons for all methods?

---

### Official Review · Reviewer_gnMm · 2025-10-29

**Soundness:** 3
**Presentation:** 3
**Contribution:** 3
**Rating:** 6
**Confidence:** 4

**Summary:**

This paper proposes MAYA, a Transformer-based architecture for tabular data that introduces two key components: the Multi-Branch of Attention (MBA) for capturing diverse intra-feature dependencies and the Inter-instance Attention Incorporated with Labels (IAIL) for modeling cross-sample relations. Empirical results show consistent improvements across various benchmarks.

**Strengths:**

- **Architectural intuition and originality**

I think the Multi-Branch of Attention (MBA) module is a well-conceived and intuitive idea. Tabular data often exhibit highly irregular and non-smooth decision boundaries—an aspect that tree-based models have long exploited through ensemble partitioning. MBA can be viewed as a soft, attention-based analogue of this principle: multiple attention branches learn complementary sub-spaces, akin to an ensemble, but within a unified Transformer framework. This design is intuitively aligned with the known structure of tabular decision boundaries.

- **Clear ablations and interpretability evidence.**

The ablation studies convincingly show how MBA and IAIL contribute independently and jointly. The accompanying visualizations help readers understand how different attention branches capture distinct feature interactions.

**Weaknesses:**

- **Lack of comparison with TabPFN.**

The experiments are extensive but omit a comparison with TabPFN, which has recently set a strong benchmark for tabular learning. Such a comparison would be helpful to establish the true empirical strength of MAYA.

- **Insufficient analysis of data-dependent behavior.**

The paper does not examine how the multi-branch attention design performs under different data conditions. Since tabular datasets vary widely in feature type composition, noise level, and sample size, these factors could strongly influence the effectiveness of MBA. A systematic study along these axes would clarify when the proposed architecture provides genuine benefits and when it may offer limited advantage.

**Questions:**

See Above.

---

### Official Review · Reviewer_go6M · 2025-10-31

**Soundness:** 2
**Presentation:** 3
**Contribution:** 2
**Rating:** 4
**Confidence:** 5

**Summary:**

his paper proposes MAYA, an encoder-decoder Transformer architecture designed to handle feature heterogeneity in tabular data.

The main contributions are twofold:
-  Encoder: A novel Multi-Branch of Attention (MBA) block is introduced. This block runs multiple parallel MHA branches and then averages their outputs, rather than concatenating them. This is intended to capture diverse feature interactions without the quadratic parameter growth seen in standard MHA-based models. The averaging is weighted using a "collaborative learning" scheme, where weights are dynamically assigned based on each branch's prediction loss.
- Decoder: An inter-instance attention mechanism, IAIL, is used. This is a cross-attention block that, during training, uses batch label embeddings as the Value. Critically, during inference, it uses the entire training set's [CLS] tokens and label embeddings as the Key and Value .

The authors evaluate MAYA on 36 datasets (14 in Part I, 22 in Part II) and claim superior performance among transformer-based methods.

**Strengths:**

- **Novel and Efficient Encoder (MBA)**: The MBA block is the paper's strongest point. It is a well-motivated and clever solution to the parameter-growth problem in standard MHA-based tabular models. The design (parallel branches with weighted averaging) is simple, effective, and well-supported by ablations and visualizations.

- **Strong Results vs. Other Transformers**: The paper is empirically rigorous within its chosen subgroup. It demonstrates SOTA performance against a comprehensive suite of other deep tabular models (AutoInt, FTT, SAINT, etc.) across 36 datasets.

- **Thorough Ablations**: The authors provide good ablation studies that validate their individual design choices for the MBA block , the IAIL decoder , and even the tokenizer's activation function.

**Weaknesses:**

- **Missing Critical Baselines (GBDTs)**: This is a fatal flaw. Any paper claiming SOTA on tabular data must compare against tuned Gradient-Boosted Decision Trees (e.g., XGBoost, CatBoost). The paper only compares against other Transformers. This is not a fair or complete comparison, and the "superior performance" claim is unsubstantiated.

- **Missing Critical Baselines (TabR)**: The IAIL decoder's inference mechanism—using the entire training set as a Key/Value store —is a kNN-like retrieval method. This is extremely similar to the mechanism in TabR (Gorishniy et al., 2024). The authors cite TabR but do not compare against it. This is a major, unacceptable omission of a highly relevant SOTA model.

- **Impractical Inference** Cost: The IAIL decoder is computationally and memory-prohibitive. Requiring L2 computation against the entire training set for every test instance is not a "limitation", it is a fundamental design flaw that makes the method non-scalable and unusable in practice.

- **Failure to Address Resubmission Feedback**: Given this is a resubmission, the failure to include GBDTs and TabR as baselines is inexplicable. This is the most basic and obvious feedback any prior reviewer would have given.

**Questions:**

- Why did you omit GBDTs (e.g., tuned XGBoost, CatBoost) from your experiments? These are the standard SOTA for tabular data. How does MAYA compare to them?

- Your IAIL decoder's inference mechanism is conceptually almost identical to the one in TabR, which you cite. Why was TabR omitted as a baseline, and how do you justify the novelty of IAIL given this prior work?

- The inference cost of IAIL seems prohibitive. Can you provide a practical wall-clock time and memory usage comparison for inference (on a single instance) against FT-Transformer or XGBoost on a medium-sized dataset (e.g., 100k training samples)?

- In your MBA branch weighting, you assign larger weights to branches with larger losses . This is counter-intuitive; one might expect to down-weight poorly performing branches. Can you provide more intuition for why this "penalizing" strategy is superior?

---

### Meta-Review · Area_Chair_t6b1 · 2026-01-06

**Summary:**

The paper provides a new transformer based architecture for tabular data. The authors devise two new components: (1) pointing out flaws in the multi head attention block, they propose Multi-Branch Attention (MBA), which computes several MHA blocks in parallel and combines their outputs using a weighted average. (2) they introduce a decoder (called IAIL) aimed to capture both intra-instance and inter-instance interactions.
The paper seems to be well written (DgCZ), and its key strength is the novel and well motivated architecture provided for the solution. Both go6M and gnMm specifically mention they were impressed by the convincing arguments leading to the design of the MBA component. DgCZ had some reservations about it, but mentioned the design as an overall strength, highlighting the second component (IAIL). In addition to the novel design, another positive aspect mentioned was the thorough ablation studies (go6M, gnMm, aEuc).

Despite these advantages, the reviews mention several flaws. The most prominent is the lack of comparison against SoTA baselines. All four reviews mentioned this issue (e.g., go6M mentions GBDT and TabR, gnMm mentioned TabPFN). The consensus over this issue shows how crucial it is to add these comparisons for a fair assessment of the method against other approaches. Another, less significant issue worth noting was the practicality of inference (mentioned by go6M and indirectly by aEuc when asking for runtime comparison).
The authors did not provide a rebuttal. In the current state the paper cannot be accepted, but it's worth mentioning that given the positive feedback for the architecture, I believe the paper does have potential, even if it doesn’t beat all SoTA models, but excels only in some particular setting. Either way, such comparisons are required for a complete paper.

**Reviewer Concerns:**

no rebuttal

**Reviewer Scores:**

no rebuttal

---

### Decision · Program_Chairs · 2026-01-26

Reject